# Unified Removal of Raindrops and Reflections: A New Benchmark and A Novel Pipeline

## Abstract

When capturing images through glass surfaces or windshields on rainy days, raindrops and reflections frequently co-occur to significantly reduce the visibility of captured images. Prior de-raindrop and de-reflection studies have failed to simultaneously remove both types of degradations from a single captured image, thereby limiting their application and robustness in real-world scenarios. In this work, we make the first attempt to explore this new task, i.e., unified removal of raindrops and reflections ($UR^3$). First of all, we set up an image acquisition platform to collect our own dataset, namely RainDrop and ReFlection (RDRF) dataset, which provides a new benchmark with substantial, high-quality, diverse image pairs. Within each pair, one has a clean foreground and the rest is corrupted by raindrops and reflections. Second, we propose a diffusion-based framework (i.e., $DiffUR^3$) to decouple the $UR^3$ task into a restoration stage and a conditional generation stage (with multiple conditions). By leveraging the powerful generative prior, $DiffUR^3$ successfully removes both degradations. Extensive experiments demonstrate that our method achieves state-of-the-art performance on our benchmark and on challenging in-the-wild images. The RDRF dataset and the codes will be made public upon acceptance.

## 1 Introduction

When we try to capture background images through raindrop-covered glasses or windscreens (a highly typical scenario is the vehicle camera recording on a rainy day), the phenomenon of reflection often coexists (see in Fig. 1 (a)). Adherent raindrops and the reflections from camera side can significantly reduce the visibility of captured images (You et al., 2016). Previously, researchers treated raindrop removal and reflection removal as two separate tasks (Qian et al., 2018; Quan et al., 2019; Shao et al., 2021; Hu et al., 2024; Zhao et al., 2025; Hu et al., 2025). Though these methods can achieve relatively good performance in removing the target type of degradation (i.e., raindrop or reflection) from a single image, they often fail to remove both types at the same time (see in Fig. 1 (c) and (d)). In this work, we aim to finding a meaningful solution capable of simultaneously eliminating raindrops and reflections, thereby enhancing the clarity of captured images. We hope this endeavor can provide support for applications such as autonomous driving, photography, and video surveillance (Zhu et al., 2025b;a).

**U**nified **R**emoval of **R**aindrops and **R**eflections ($UR^3$) is a fundamental but complex task. Its key challenges lie in the following three aspects. (1) Lack of data: deep learning based methods requires a large number of image pairs for training process. Currently, no publicly available dataset exists wherein the low-quality images contain both raindrop degradation and reflection degradation. (2) Task gap: certain gap exists between raindrop removal and reflection removal. Cross-task adaptation of pretrained models often yields sub-optimal performance due to domain shift. (3) Void information: in regions with exceptionally large/dense raindrops or intense reflections, background scene is completely lost. These occluded regions are extremely challenging, somewhat analogous to the inpainting task.

Trying to address these challenges, we first set up an image acquisition platform to collect corresponding data for the $UR^3$ task. We collect a substantial number of image pairs to constitute our **R**ain**D**rop and **R**e**F**lection (RDRF) dataset, wherein one image features a clean foreground, while the other is a degraded image containing both raindrops and reflections (both images share identical

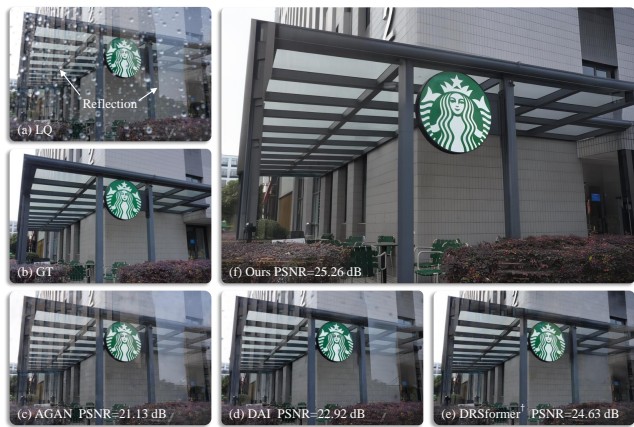

Figure 1: (a) A low-quality image with raindrops and reflections. (b) The ground truth. (c) The processed result of AGAN (Qian et al., 2018). (d) The processed result of DAI (Hu et al., 2025). (e) The processed result of a re-trained restoration method (Chen et al., 2023b). Superscript † means re-trained. (f) The processed result of our DiffUR$^3$ pipeline.

background). We hope RDRF dataset can contribute to the advancement and development of the UR$^3$ task, and benefit the entire community.

Then, we propose a diffusion-based framework (i.e., DiffUR$^3$) based on IRControlNet (Lin et al., 2024) to jointly remove these two kinds of degradations. Within this framework, the UR$^3$ task is decoupled into two stages: (I) Restoration: remove some simple degradations without introducing artifacts to offer a reliable condition image. (II) Conditional generation: guided by **multiple** condition images, reconstruct the challenging regions by leveraging the powerful generative prior.

Last but not least, we design a Modulate&Gate module to align each condition with the noisy latent and adaptively select the beneficial components in the latent space. We also train an additional fidelity encoder to offer faithful features for guiding VAE Decoder to maintain the texture and structure of the reconstructed image.

## 2 RELATED WORK

### 2.1 RAINDROP REMOVAL

In the realm of raindrop removal, recent studies have explored diverse methodologies. Eigen et al. (2013) pioneered single-image raindrop removal using CNNs. Qian et al. (2018) introduced a generative adversarial network (GAN) to enhance raindrop removal. Transformer-based approaches like IDT (Xiao et al., 2023), UDR-S$^2$Former (Chen et al., 2023a) and Histoformer (Sun et al., 2024) have demonstrated superior performance. Meanwhile, the CCN (Quan et al., 2021) adopts a unique approach by employing neural architecture search. More recently, diffusion-based methods like WeatherDiff (Özdenizci & Legenstein, 2023) and T$^3$-DiffWeather (Chen et al., 2024) have emerged, leveraging the generative capabilities of diffusion models to enhance raindrop removal.

### 2.2 REFLECTION REMOVAL

In the field of single image reflection removal, various advanced techniques have been proposed to address the ill-posed nature of separating superimposed transmission and reflection layers. Early methods such as CEILNet (Fan et al., 2017) leverage edge information and deep learning. IBCLN (Li et al., 2020) introduces a cascaded refinement strategy to iteratively enhance the estimates of transmission and reflection layers. More recent advancements, YTMT (Hu & Guo, 2021), DSRNet (Hu & Guo, 2023) and DSIT (Hu et al., 2024), employ dual-stream networks to enhance feature interaction and correlation assessment. Further more, diffusion-based models like L-DiffER (Hong et al., 2024) and DAI (Hu et al., 2025) show impressive removal capabilities across a wide range of real-world scenarios.

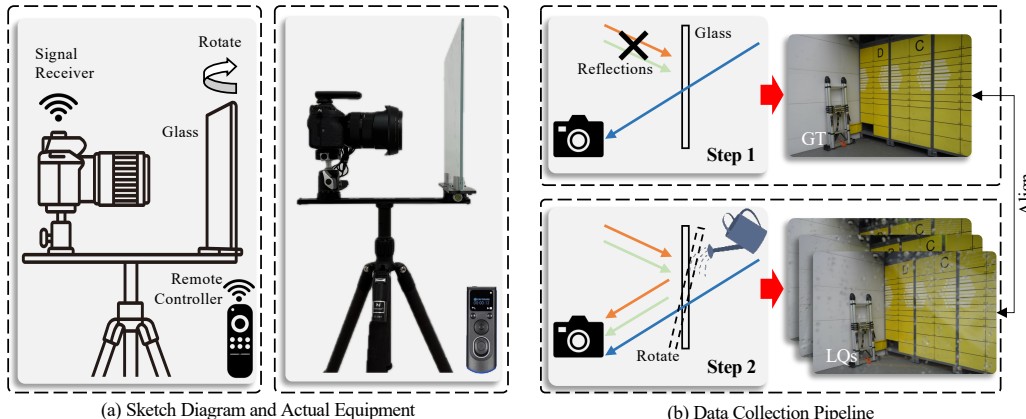

(a) Sketch Diagram and Actual Equipment

(b) Data Collection Pipeline

Figure 2: (a) Sketch diagram and actual equipment of our image acquisition platform. To suppress shutter-induced micro-vibrations which may potentially induce image misalignment, we implement a wireless triggering mechanism. It comprises a remote controller and a camera-mounted signal receiver, enabling contact-free shutter operation. (b) The data collection pipeline for our RDRF dataset. ✖ denotes light occlusion

## 2.3 DATASETS

Existing real-world datasets for raindrop removal include AGAN (Qian et al., 2018), RainDS (Quan et al., 2019), RobotCar (Porav et al., 2019), and Raindrop Clarity (Jin et al., 2024), these datasets provide low-quality images with raindrops and their corresponding ground truth images. Differently, Windshield (Soboleva & Shipitko, 2021) contains degraded images along with their corresponding binary masks that indicate the raindrop-affected areas. For reflection removal task, it is noteworthy that synthetic data is commonly employed for training. Recently, some real-world datasets have been proposed, such as RRW (Zhu et al., 2024) and DRR (Hu et al., 2025). However, there are no existing datasets that specifically address the unified removal of raindrops and reflections, which is the focus of our work.

## 3 RDRF DATASET

Similar to most deep learning based methods, our task (i.e., unified removal of raindrops and reflections) requires a large number of degraded images with corresponding clean labels for training. There are no existing training or testing datasets for this new task. As shown in Fig. 2 (a), we set up an image acquisition platform to collect our own **R**ain**D**rop and **R**e**F**lection (RDRF) dataset. In our case, a substantial volume of image pairs are required, where each pair comprises two images with the identical background scene, yet one has a clean foreground and the other is corrupted by raindrops and reflections.

### 3.1 HARDWARE

Drawing inspirations from previous works of Zhu et al. (2024) and Li et al. (2024), our RDRF dataset is captured under real scenarios deliberately constructed in controlled environments. For the hardware configuration, the camera is mounted on a tripod using an adjustable base, with the glass slab positioned in front of the lens. We connect a signal receiver onto the camera, thereby enabling remote control of the shutter. This wireless triggering mechanism can effectively avoid image misalignment caused by camera vibrations resulting from manual operation. To ensure diversity, neither the camera nor the glass is fixed. They can be adjusted to simulate different shooting situations (e.g., camera-to-glass distances/angles). In addition, we utilize two cameras (Sony ILCE-7RM4A and Nikon D7100) with zoom lens and choose different glass thicknesses (3 mm, 5mm, and 8 mm) to further enhance diversity.

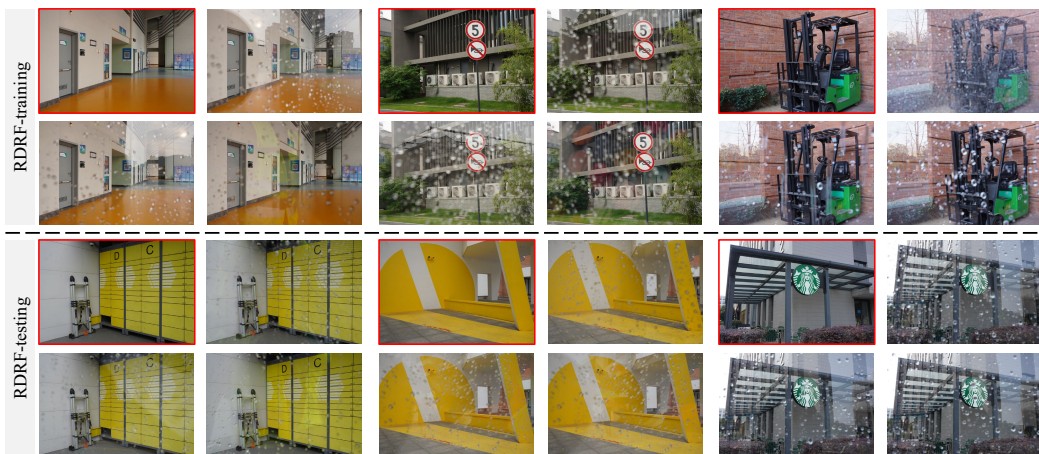

Figure 3: Our RDRF dataset comprises a diverse collection of scenes, each contains a ground truth and multiple low-quality images. As illustrated in this figure, the clean ground truths are highlighted in red boxes, while corresponding low-quality images are arranged around. We divide it into the training and testing subsets, ensuring no overlapping samples between them. Please check and zoom in on screen for a better view.

## 3.2 DATA COLLECTION PIPELINE

Fig. 2 (b) exhibits our data collection pipeline. For step 1, we utilize a light-blocking box to suppress the reflections from the camera side (✖ denotes light occlusion). The obtained image is regarded as the ground-truth. For step 2, we keep the background scenario and camera unchanged. The light-blocking box is removed and the raindrops are created by spraying water onto the glass surface. By randomly rotating the glass at different angles, we create varying reflections with different scenes and intensities. For each scene, multiple images are captured as the low-quality ones.

Some samples are illustrated in Fig. 3. As demonstrated in the dataset, our RDRF dataset comprises a comprehensive collection of scenes. The raindrops are captured under diverse shapes and sizes (circular, elliptical, and irregular), ranging from sparse to dense. Raindrop flow traces are also included. In addition, the reflections are also captured with diverse reflection scenes, ranging from weak to strong. All the images are captured in $4752 \times 3168$ resolution to ensure high-quality.

In total, our RDRF dataset consists of 252 unique scenes (we categorize these scenes into eight distinct classes, namely: building facades & structures, streets & traffic, public & open spaces, functional components of building, industrial & commercial facilities, signs & markers, infrastructure & obstacles, and others.). **The category distribution diagram can be found in the Appendix A.1 (i.e., Fig. 10).** It is divided into a training set (216 scenes with 9003 image pairs) and a testing set (36 scenes with 83 image pairs). There are no overlapping samples between them.

To address the spatial misalignment caused by our hardware, we follow the procedures proposed in (Wan et al., 2017). It starts by extracting SIFT (Lowe, 2004) keypoints and descriptors, which are matched with L2 distance. Using the matched keypoints, a homography matrix is estimated via RANSAC (Fischler & Bolles, 1981) to handle outliers and find a robust geometric transformation. The low-quality image is aligned to the ground truth by applying a perspective warp using the computed homography.

Our RDRF dataset represents the first-of-its-kind contribution to the UR[3] task. Some researches have also noticed the reflection artifacts in the early raindrop datasets (Qian et al., 2018; Quan et al., 2019) and proposed a highly valuable dataset to address the raindrop removal task only (i.e., Raindrop Clarity (Jin et al., 2024)). Their brilliant dataset has different focus with ours.

## 4 METHODOLOGY

Our RDRF dataset provides sufficient and diverse training data for UR[3] task. Formally, given a low-quality (LQ) image $I_{lq} \in \mathbb{R}^{3 \times H \times W}$ with both raindrops and reflections on it, a straightforward

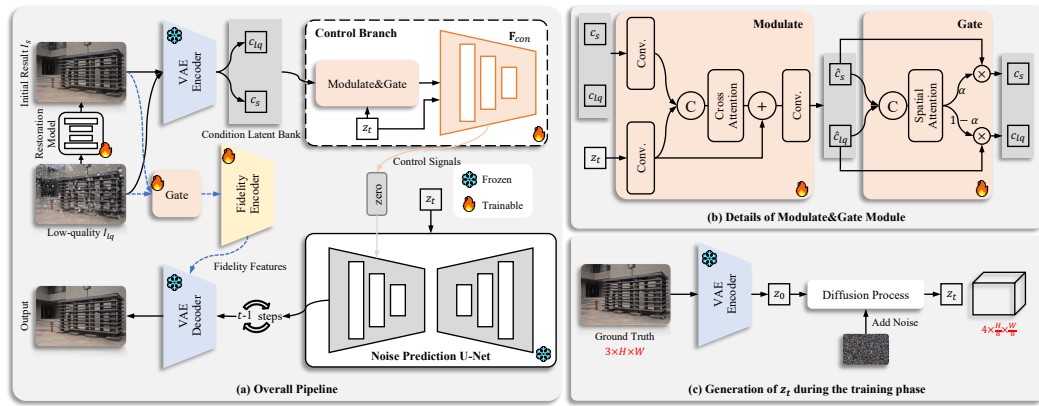

Figure 4: (a) Overall pipeline of our DiffUR[3] framework. It consists of a restoration stage and a conditional generation stage. Given a low-quality image $I_{lq}$, the restoration stage removes the undesired degradation to obtain the initial result $I_s$. Both $I_{lq}$ and $I_s$ are fed into the next stage as the condition images. We inject the condition information through a control branch, which outputs control signals for the noise prediction U-Net. (b) Details of the Modulate&Gate module within the control branch. (c) The generation of noisy latent $z_t$ during the training phase. Note that the noisy latent starts from random Gaussian noise during the inference.

idea is to employ conventional restoration methods (Chen et al., 2023b; Sun et al., 2024) to directly learn the mapping function from the low-quality to the ground-truth. However, their results are perceptually unsatisfying, because of the complexity of UR[3] task. An example can be found in Fig. 1 (e). Instead, we try to utilize the powerful generative priors of the diffusion model as an effective solution. In this way, UR[3] is regarded as a conditional image generation problem.

By referring to DiffBIR (Lin et al., 2024) framework, we design a two-stage network (i.e., DiffUR[3]) shown in Fig. 4 (a), which also consists of **(I) a restoration stage**, and **(II) a conditional generation stage**. The restoration stage outputs an initial result $I_s$, which is then used as the condition image in the following generation stage.

Although similar to DiffBIR framework, our DiffUR[3] has two major different designs: (1) Considering the fact that raindrops and reflections merely affect certain regions, some parts in the LQ image can be regarded as clean. We also integrate the LQ image $I_{lq}$ as one of the conditions during stage II. (2) We employ a fidelity encoder to provide faithful structural information for the decoding process. Details of our DiffUR[3] pipeline are described below.

## 4.1 RESTORATION STAGE

In the first stage, our aim is to remove some simple yet undesired degradations from LQ input $I_{lq}$ without introducing artifacts. The output image $I_s$ provides a reliable condition image for training the generation stage.

$$I_s = \mathcal{RM}(I_{lq}), \tag{1}$$

where $\mathcal{RM}(\cdot)$ denotes the restoration model. In our implementation, we select the DRSformer (Chen et al., 2023b) as the restoration model in stage I due to its superior performance and generalization capability.

## 4.2 CONDITIONAL GENERATION STAGE

Our generation stage is based on the Stable Diffusion Model (Rombach et al., 2022), because its powerful generative prior can facilitate the restoration of regions that are challenging to be recovered in stage I through a conditional generation approach. To achieve better efficiency and stabilized training, the pretrained VAE (Kingma & Welling, 2013) encoder $\mathcal{E}$ is employed to encode the condition images into the latent space. Both diffusion and denoising processes are performed in this

space instead of the pixel space. The main denoising network is a pretrained U-Net. The denoising output is then converted back to the pixel space using the pretrained VAE decoder $\mathcal{D}$.

As mentioned above, for UR$^3$ task we argue LQ image $I_{lq}$ contains clean information within certain regions [1]. Therefore, both $I_{lq}$ and $I_s$ are encoded by $\mathcal{E}$:

$$c_{lq}, c_s = \mathcal{E}(I_{lq}, I_s), \tag{2}$$

where $c_{lq} \in \mathbb{R}^{4 \times \frac{H}{8} \times \frac{W}{8}}$ and $c_s \in \mathbb{R}^{4 \times \frac{H}{8} \times \frac{W}{8}}$ denote the obtained condition latent from $I_{lq}$ and $I_s$, respectively. Besides, the noisy latent $z_t$ is also embedded, since it has been proven to enhance image quality (Lin et al., 2024). The generation of $z_t$ is shown in Fig. 4 (c).

Similar to previous work (Lin et al., 2024), we also inject the condition information via a control branch. We make a trainable copy of the pretrained U-Net encoder and middle block (i.e., $\mathbf{F}_{con}$ in Fig. 4 (a)), which receives condition information and then outputs control signals. A normal solution is to add or concatenate $c_{lq}$, $c_s$ and $z_t$ before sending to $\mathbf{F}_{con}$ (Chen et al., 2025; Özdenizci & Legenstein, 2023). However, we observe that the noisy latent $z_t$ varies at different time steps, yet the condition latent (i.e., $c_{lq}$ or $c_s$) remains unchanged. Instead of direct addition or concatenation, we propose a more reasonable solution to modulate $c_{lq}$ and $c_s$ through $z_t$. Since there are more than one condition latent, a gate mechanism is introduced to adaptively assign different spatial weights to $c_{lq}$ and $c_s$.

To this end, before entering the $\mathbf{F}_{con}$, we design a Modulate&Gate module which consists of a Modulate block and a Gate block. Fig. 4 (b) shows the details of our Modulate&Gate module. We describe them as below.

### 4.2.1 MODULATE BLOCK

Take $c_s$ as an example, $c_{lq}$ can be similarly derived. First, both $c_s$ and $z_t$ individually pass through a convolutional layer to extract their features $f_c \in \mathbb{R}^{C \times \frac{H}{8} \times \frac{W}{8}}$ and $f_z \in \mathbb{R}^{C \times \frac{H}{8} \times \frac{W}{8}}$. $C$ denotes the channel number of the extracted feature. In our implementation, we set $C = 32$. Then, their concatenation result is fed into two consecutive transformer layers (Vaswani et al., 2017) to perform the cross attention operation, which can facilitate the information interaction between $f_c$ and $f_z$. Our cross attention operation aligns the dimensions of the output $f_{cross}$ and $f_z$ at the end. Finally, we add $f_{cross} \in \mathbb{R}^{C \times \frac{H}{8} \times \frac{W}{8}}$ with $f_z$, and employ another convolutional layer to reduce the channel number back to $4$. The formulations are as follows:

$$\begin{aligned} f_c, f_z &= Conv(c_s, z_t), \\ f_{cross} &= CrAttn([f_c, f_z]), \\ \hat{c}_s &= Conv(f_{cross} + f_z), \end{aligned} \tag{3}$$

where $Conv(\cdot)$ denotes the convolutional layer, $CrAttn(\cdot)$ denotes the cross attention operation, $[\cdot, \cdot]$ denotes the concatenation, $\hat{c}_s$ is the modulated condition latent, and $\hat{c}_{lq}$ can be derived by replacing $c_s$ with $c_{lq}$ in Eqn. 3.

### 4.2.2 GATE BLOCK

After obtaining $\hat{c}_s$ and $\hat{c}_{lq}$, we need to selectively extract the components that are beneficial to our DiffUR$^3$. We concatenate them together, and then send to a spatial attention to generate a spatial weight $\alpha \in \mathbb{R}^{\frac{H}{8} \times \frac{W}{8}}$. The spatial attention operation consists of two convolutional layers, one activation layer, and one sigmoid layer. The formulations are as follow:

$$\begin{aligned} \alpha &= SpAttn([\hat{c}_s, \hat{c}_{lq}]), \\ \bar{c}_s &= \alpha \cdot \hat{c}_s, \\ \bar{c}_{lq} &= (1 - \alpha) \cdot \hat{c}_{lq}, \end{aligned} \tag{4}$$

where $SpAttn(\cdot)$ denotes the spatial attention operation, $\bar{c}_s$ and $\bar{c}_{lq}$ are the output condition latent variables. Note that our Modulate&Gate module is simple yet effective. More sophisticated designs can be considered for better performance, which is not the focus of this work.

---

[1]Unlike blind super-resolution and blind image denoising in (Lin et al., 2024), where the entire LQ image is degraded.

We concatenate $\bar{c}_s$, $\bar{c}_{lq}$, and the noisy latent $z_t$ together, and send them to $\mathbf{F}_{con}$ for generating the control signals, which are added to the denoising U-Net via zero convolutions (Zhang et al., 2023). At each time step, the noise prediction U-Net estimates the noise component and performs denoising on the noisy latent $z_t$. During the inference phase, the noisy latent starts from random Gaussian noise and iteratively passes through the pretrained U-Net to estimate the clean latent $\hat{z}_0$.

### 4.3 ADDITIONAL FIDELITY ENCODER

Even with the control branch, we still observe some unwanted textural and structural distortions in some cases after decoding estimated clean latent $\hat{z}_0$ back to pixel space. An example can be found in Fig. 7 (a). To deal with this issue and improve the fidelity of generated results, we train a fidelity encoder after the training of control branch, inspired by (Chang et al., 2023). Fig. 5 is the training pipeline of our fidelity encoder, which shares the same architecture with VAE encoder (Kingma & Welling, 2013) (besides the first convolutional layer).

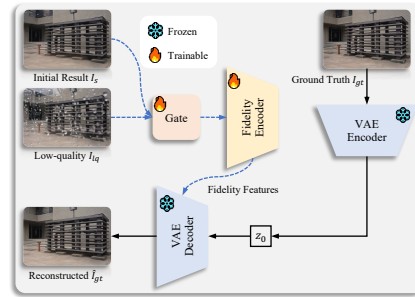

To keep consistent with the control branch, both LQ image $I_{lq}$ and initial result $I_s$ are fed into the fidelity encoder through a Gate block to extract faithful features. The extracted fidelity features are added to corresponding positions in VAE Decoder via zero convolutions. Then, we

Figure 5: Training pipeline of our fidelity encoder.

encode the ground truth $I_{gt}$ via the pretrained VAE encoder to latent space, simulating the denoised latent i.e., $z_0$. Finally, guided by the fidelity features, pretrained VAE decoder (Kingma & Welling, 2013) converts the compressed latent $z_0$ to a reconstructed image $\hat{I}_{gt}$. The whole pipeline is trained by minimizing a mean absolute error (i.e., $L_1$ loss) between $\hat{I}_{gt}$ and $I_{gt}$.

## 5 EXPERIMENTAL RESULTS

### 5.1 IMPLEMENTATION DETAILS AND METRICS

Our DiffUR[3] is trained on our RDRF-training dataset. We train the restoration model in stage I for 300k iterations (batch size = 4) on a single RTX 4090 GPU. Then we adopt the Stable Diffusion 2.1-base (Rombach et al., 2022) as the generative prior, and train the control branch in stage II for 50k iterations (batch size = 40) on two A6000 GPUs. The fidelity encoder is trained for 300k iterations (batch size = 6) on two A6000 GPUs. **More details can be found in Appendix A.2.**

We adopt three traditional metrics (PSNR, SSIM, LPIPS (Zhang et al., 2018)) and four no-reference image quality assessment metrics (MUSIQ (Ke et al., 2021), CLIPIQA (Wang et al., 2023), CLIP-IQA+ (Wang et al., 2023), HyperIQA (Su et al., 2020)) to evaluate our performance.

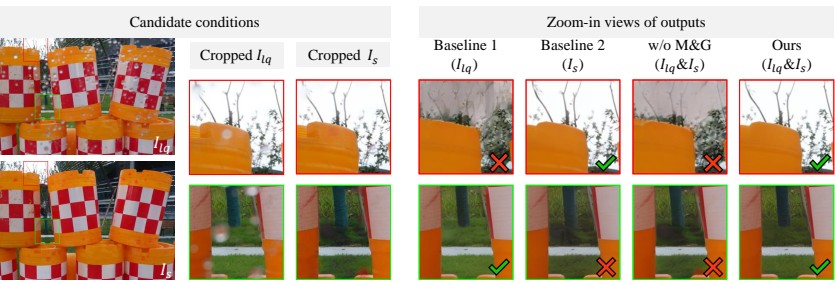

Figure 6: Ablation study of our Modulate&Gate (M&G) module. ✔ and ✘ denote good and bad results, respectively.

## 5.2 ABLATION STUDY

### 5.2.1 OVERALL

First, we employ the naive diffusion-based method which solely adopts $I_{lq}$ as the condition image (similar to (Özdenizci & Legenstein, 2023)) and denote it as **Baseline 1**. In addition, **Baseline 2** means only the $I_s$ is regarded as the condition image (similar to (Lin et al., 2024)). To validate the effectiveness of key components in our DiffUR[3], we perform ablation studies of modulate&gate module and fidelity encoder on

Table 1: Ablation study of key components.

| Model | PSNR↑ | SSIM↑ | LPIPS↓ |
|---|---|---|---|
| Baseline 1 ($I_{lq}$ as condition) | 28.84 | 0.9302 | 0.0817 |
| Baseline 2 ($I_s$ as condition) | 29.18 | 0.9311 | 0.0838 |
| w/o Modulate&Gate Module | 29.21 | 0.9322 | 0.0802 |
| w/o Fidelity Encoder | 27.64 | 0.8198 | 0.0990 |
| DiffUR[3] | 29.84 | 0.9400 | 0.0733 |

our RDRF-testing dataset. We remove corresponding components from our DiffUR[3] and denote them as '**w/o components**' in Table 1.

Table 1 summarize the quantitative results in terms of PSNR, SSIM, and LPIPS. We observe that both modulate&gate module and fidelity encoder are critically important for our DiffUR[3], as omitting either component leads to a significant performance degradation.

### 5.2.2 MODULATE&GATE MODULE

In Table 1, w/o Modulate&Gate Module means both $I_{lq}$ and $I_s$ are embedded as the condition images and fused by channel-wise concatenation in latent space. We provide an in-depth analysis on the function of modulate&gate module. As illustrated in Fig. 6, Baseline 1 occasionally exhibits generation errors due to the inherent characteristics of diffusion model. In contrast, Baseline 2 relies on the condition image $I_s$. They demonstrate distinct advantages across different regions. Simple channel-wise concatenation fails to systematically integrate their complementary strengths (third column). The introduced modulate&gate module enables adaptive integration of information from dual condition images, thereby enhancing the model performance.

### 5.2.3 FIDELITY ENCODER

We also provide the qualitative analysis on the function of fidelity encoder. As shown in Fig. 7, when the fidelity encoder is excluded, the recovered text exhibits distortions (which is highly difficult to be recognized by human begins), and some structure deformations emerge in the scene.

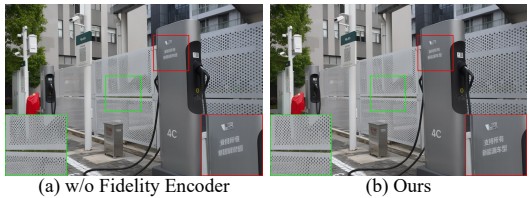

Figure 7: Ablation study of our Fidelity Encoder.

Figure 8: User study on RDRF-testing.

## 5.3 COMPARISONS WITH STATE-OF-THE-ART METHODS

Since this work is the first exploration for unified removal of raindrops and reflections (UR[3]) task. There are no prior methods. We employ three classical raindrop removal methods (i.e., AGAN (Qian et al., 2018), Histoformer (Sun et al., 2024), WeatherDiff[64] (Özdenizci & Legenstein, 2023)), three classical reflection removal methods (i.e., RDNet (Zhao et al., 2025), DSIT (Hu et al., 2024), DAI (Hu et al., 2025)), two cascaded methods (de-raindrop then de-reflection, and de-reflection then de-raindrop) as the competitors. We adopt their published models for these methods. In addition, by utilizing their published codes, we re-train AGAN, DRSformer, Histoformer, RaindropDiff[64] on our RDRF-training dataset, endowing them with capabilities for UR[3] task. We do not re-train reflection removal models because they typically require access to additional synthetic data for training, which would make the comparison unfair.

Table 2: Benchmark results on our RDRF-testing dataset. We report PSNR, SSIM, LPIPS and four no-reference image quality assessment metrics (i.e., MUSIQ, CLIPIQA, CLIPIQA+, HyperIQA) to perform comprehensive comparisons. The **bold** and underline indicate the best and second best.

| Type | Method | RDRF-testing | | | | | | |
| --- | --- | --- | --- | --- | --- | --- | --- | --- |
| | | PSNR↑ | SSIM↑ | LPIPS↓ | MUSIQ↑ | CLIPIQA↑ | CLIPIQA+↑ | HyperIQA↑ |
| Raindrop removal | AGAN | 25.71 | 0.9258 | 0.0990 | 74.36 | 0.4844 | 0.6479 | 0.6915 |
| | Histoformer | 26.47 | 0.9300 | 0.0990 | 74.08 | 0.4315 | 0.6226 | 0.6816 |
| | WeatherDiff$_{64}$ | 25.19 | 0.9067 | 0.1074 | 73.85 | 0.4497 | 0.6604 | 0.6679 |
| Reflection removal | RDNet | 27.11 | 0.9232 | 0.1076 | 72.34 | 0.4637 | 0.6243 | 0.6585 |
| | DSIT | 26.53 | 0.9207 | 0.1130 | 72.86 | 0.4450 | 0.6326 | 0.6585 |
| | DAI | 27.75 | 0.9294 | 0.0951 | 74.53 | 0.4556 | 0.6413 | 0.6896 |
| Cascaded | Histoformer+DAI | 28.13 | 0.9395 | 0.0804 | 75.31 | 0.4743 | 0.6490 | 0.7083 |
| | DAI+Histoformer | 27.78 | 0.9340 | 0.0851 | 75.09 | 0.4768 | 0.6520 | 0.7007 |
| Re-trained | AGAN$^\dagger$ | 26.02 | 0.9145 | 0.1475 | 71.85 | 0.4193 | 0.5768 | 0.6669 |
| | Histoformer$^\dagger$ | 29.53 | 0.9384 | 0.0745 | 73.38 | 0.4738 | 0.6433 | 0.6764 |
| | RaindropDiff$_{64}$$^\dagger$ | 26.95 | 0.9236 | 0.0922 | 73.49 | 0.5013 | 0.6662 | 0.6493 |
| | DRSformer$^\dagger$ | 29.61 | **0.9417** | 0.0745 | 74.16 | 0.4706 | 0.6536 | 0.6864 |
| Ours | DiffUR$^3$ | **29.84** | 0.9400 | **0.0733** | **75.41** | **0.5018** | **0.6800** | **0.7256** |

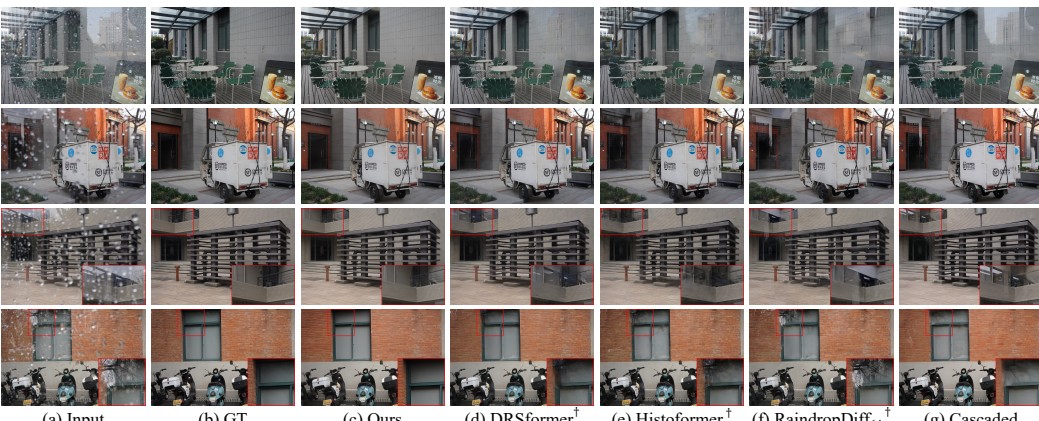

(a) Input  (b) GT  (c) Ours  (d) DRSformer$^\dagger$  (e) Histoformer$^\dagger$  (f) RaindropDiff$_{64}$$^\dagger$  (g) Cascaded

Figure 9: Visual results of various methods on our RDRF-testing. For Cascaded in (g), we choose Histoformer+DAI. Superscript $\dagger$ means this method is re-trained on our RDRF-traing dataset. Please check and zoom in on screen for a better view.

Table 2 shows the quantitative results on our RDRF-testing dataset. Note that our DiffUR$^3$ ranks the first among six metrics, except SSIM. We conduct a user study to evaluate our DiffUR$^3$ subjectively against other methods. The statistical results in Fig. 8 indicates that our DiffUR$^3$ is more favored by the invited experts. **More details about the user study can be found in Appendix A.3.** In addition, some visual comparisons of our DiffUR$^3$ and the competitors are provided in Fig. 9. It is worth mentioning that the results of our DiffUR$^3$ are closer to the ground truth with less degradation residuals and artifacts than the alternatives. We also capture some testing images from real-world driving scenarios in rainy weather to form the RDRF-wild dataset. **More visual results of RDRF-testing and RDRF-wild can be found in Appendix A.4.**

## 6 CONCLUSION

This work introduces a pioneering approach to the challenging task of UR$^3$. By establishing the first dedicated dataset (RDRF dataset) and proposing a novel diffusion-based framework (DiffUR$^3$), we successfully address the limitations of previous methods that treat raindrop and reflection removal as separate tasks. Our two-stage pipeline, incorporating a restoration stage and a conditional generation stage, effectively leverages generative priors to remove both types of degradations simultaneously. Extensive experiments demonstrate the superiority of our approach over state-of-the-art methods. The RDRF dataset and DiffUR$^3$ framework contribute significantly to the advancement of the UR$^3$ task, offering valuable resources for future research.

## ETHICS STATEMENT

We have carefully reviewed and adhered to the ICLR Code of Ethics. Our research does not involve human subjects, personal data, or sensitive attributes, and thus does not raise concerns regarding privacy or informed consent. We declare that there are no conflicts of interest, commercial sponsorships, or ethical concerns beyond those stated above.

## REPRODUCIBILITY STATEMENT

We have taken several measures to ensure the reproducibility of our results. The main paper and the appendix provide detailed descriptions of the model architecture, training procedure & hyperparameters, and evaluation protocols.

To further facilitate reproducibility, We provide the essential code snippets in the supplementary material to illustrate the core components of our method. In addition, the random seed used in all experiments is fixed, ensuring consistent results. These efforts are intended to make it possible for other researchers to replicate and extend our findings.

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

# A APPENDIX

## USE OF LLMS

In accordance with ICLR's guidelines, we disclose that large language models (LLMs) were employed solely as a general-purpose tool for aiding or polishing writing in this submission.

### A.1 CATEGORY DISTRIBUTION DIAGRAM

Our RDRF dataset covers a diverse range of urban scene categories with varying semantic attributes. As shown in Fig. 10, the category distribution diagram reveals a rich diversity of semantic groups, along with a relatively balanced representation across them, thereby providing a solid foundation for the $UR^3$ task.

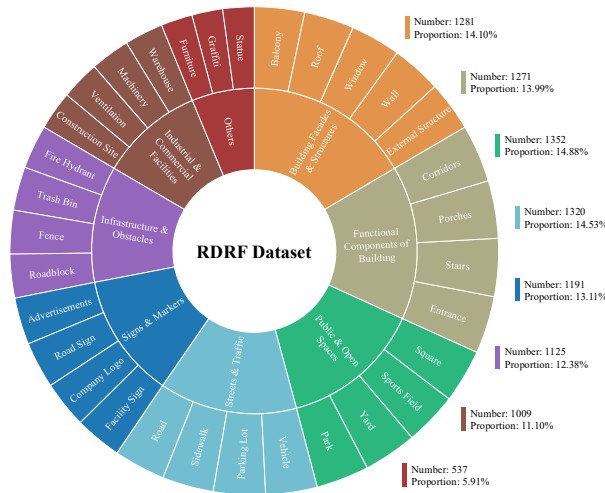

Figure 10: The category distribution diagram of our RDRF dataset.

### A.2 IMPLEMENTATION DETAILS

For the training of stage I, we adopt DRSformer (Chen et al., 2023b) as the restoration model. $\{N_0, N_1, N_2, N_3, N_4\}$ are set to $\{0,2,4,4,8\}$, and the number of attention heads for sparse transformer blocks (i.e., STBs) in level 1 to level 4 are set to $\{1,2,4,8\}$. The initial channel $C$ is set to 16. Compared with the original, we make certain simplifications to accelerate the computation in Stage I. Patches of size $256 \times 256$ are randomly cropped from the RDRF-training dataset, and horizontal and vertical flips are applied as the data augmentation techniques. This model is trained with a batch size of 4 and with an initial learning rate of $3e^{-4}$ for the first 100K iterations, which will gradually reduced to $1e^{-6}$ using cosine annealing schedule (He et al., 2019) during the remaining 200K iterations.

For stage II, the images in our RDRF-training dataset are randomly cropped into $512 \times 512$ patches. Horizontal and vertical flips, resizing, rotation are applied as the data augmentation techniques. The control branch is trained with a batch size of 40 and with a fixed learning rate of $1e^{-4}$ for entire 50K iterations. To accelerate the sampling process, we adopt a spaced DDPM sampling schedule (Nichol & Dhariwal, 2021) which requires 50 sampling steps.

For the training of the additional fidelity encoder, we follow the training settings of stage II, except the batch size and number of iterations.

Note that the images in our RDRF-training dataset are firstly resized to a fixed resolution of $1080 \times 720$, and we use AdamW (Loshchilov et al., 2017) optimizer with default settings for all the training procedures.

## A.3 DETAILS OF OUR USER STUDY

We conduct a user study to evaluate our DiffUR$^3$ subjectively against five methods (i.e., DRSformer (Chen et al., 2023b), Histoformer (Sun et al., 2024), Histoformer+DAI (Sun et al., 2024; Hu et al., 2025), RaindropDiff$_{64}$ (Özdenizci & Legenstein, 2023), AGAN (Qian et al., 2018)). Apart from the cascaded method (Histoformer+DAI), all the other methods are re-trained on our RDRF dataset. Specifically, we randomly select 50 images from our RDRF-testing dataset and invite 20 experts with image restoration background as volunteers. For every image, each expert is asked to compare the result of our DiffUR$^3$ with the alternatives one by one. For each comparison, the observers are demanded to choose the favored one after at least 10 seconds of observation. Afterward, we statistic the percentage of certain method to be selected.

## A.4 ADDITIONAL VISUAL RESULTS OF RDRF-WILD AND RDRF-TESTING

Images collected in real-world scenarios often exhibit lower quality than those captured in controlled environments, making them more suitable for evaluating model robustness in practical settings. We capture a set of images from actual driving environments, namely **RDRF-wild**. As shown in Fig. 11, the results demonstrate that our method effectively removes raindrops and reflections, significantly enhancing image quality. This indicates the robustness and generalization capability of our approach in real-world applications. We also provide additional visual results of various methods on our RDRF-testing in Fig. 12.

## A.5 DOWNSTREAM APPLICATION

To further demonstrate the downstream applicability of our DiffUR$^3$, we employ Google Vision API [2] to test whether our outputs can improve the object detection performance. Specifically, we conduct object detection on degraded images and our restored results. As shown in Fig. 13, the original inputs yield missing or incorrect object annotations due to the visual interference of raindrops and reflections. In contrast, our DiffUR$^3$ effectively assist the detector in recognizing the omitted objects. This proves that our method not only improves perceptual image quality but also enhances performance in high-level vision tasks, validating its potential for real-world applications.

## A.6 DISCUSSION ON STAGE I

Our DiffUR$^3$ pipeline represents a highly flexible framework. Although it needs a condition image generated by the restoration model (i.e., DRSformer) within stage I during the training phase, this condition can be discarded during the testing phase, and instead, another condition image generated by other models (e.g., Histoformer (Sun et al., 2024), DAI (Hu et al., 2025)) can be utilized. This indicates that the DiffUR$^3$ has already learned the capability to extract valuable information from the condition image. Fig. 14 shows the experimental results with different stage I. By replacing DRSformer with DAI during the inference phase, our DiffUR$^3$ still can output pleasing result (Fig. 14 (c)). Note this DAI model has not been re-trained on our RDRF dataset. We also extend our DiffUR$^3$ to let it incorporate two stage I (Fig. 14 (d)). This model achieves the best performance. We will further delve into an in-depth exploration of this research direction.

## A.7 GENERALIZATION TO SINGLE-DEGRADATION DATASET

We also apply our DiffUR$^3$ on raindrop-only dataset (i.e., Qian's dataset (Qian et al., 2018)). It contains 861 image pairs for training and another 58 pairs for testing. The benchmark results on the testing image pairs (i.e., Test-a) are listed in Table 3. We observe that our DiffUR$^3$ ranks first in terms of no-reference image quality assessment metrics, which aligns closely with human visual perception (some visual results are illustrated in Fig. 15). It should be noted that Qian's dataset has a relatively small number of samples (861 pairs) and relatively low resolutions (mostly merely $720 \times 480$), which is detrimental to the training of diffusion models and may lead to sub-optimal results. During the training, we have to reduce the patch size to accommodate Qian's dataset.

---

[2]https://cloud.google.com/vision.

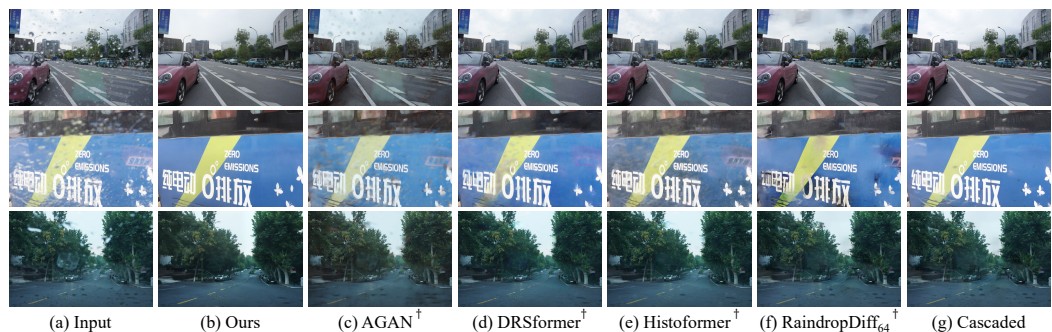

Figure 11: Visual results of various methods on our RDRF-wild dataset. Please check and zoom in on screen for a better view.

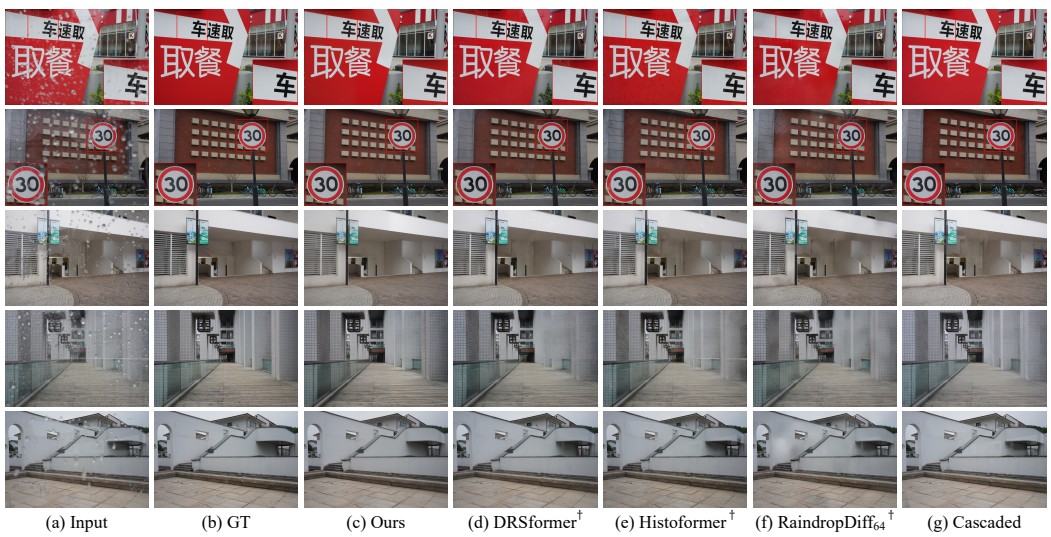

Figure 12: More visual results of various methods on our RDRF-testing dataset. Please check and zoom in on screen for a better view.

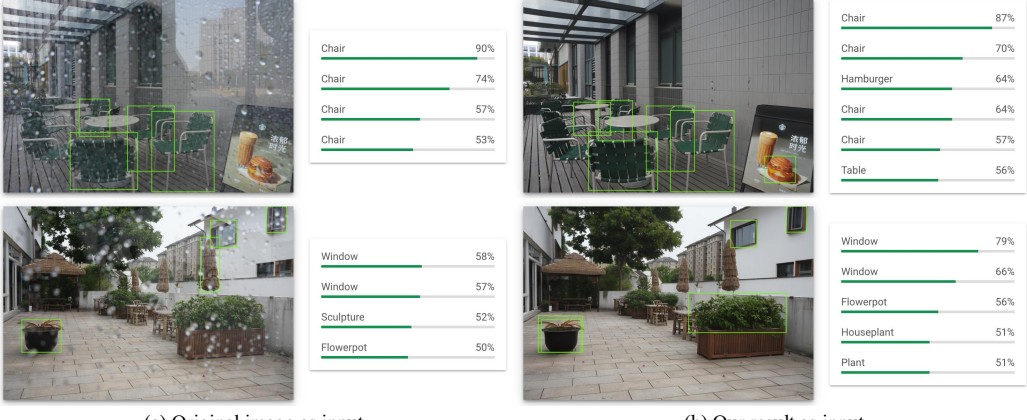

Figure 13: We conduct object detection on original images and our restored results.

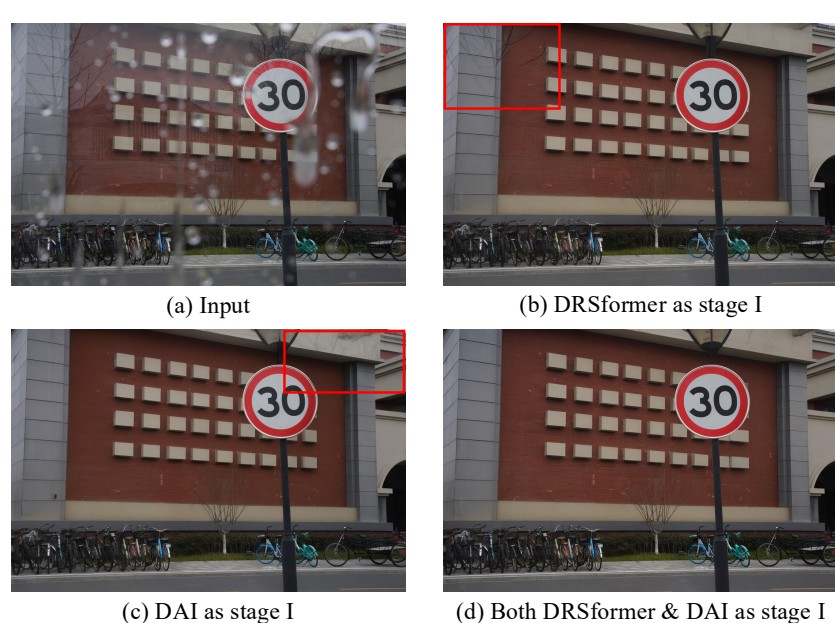

Figure 14: Experimental results with different settings of stage I.

Table 3: Benchmark results on Test-a dataset. We report PSNR, SSIM and three no-reference image quality assessment metrics (i.e., MUSIQ, CLIPIQA+, HyperIQA) to perform comprehensive comparisons. The **bold** and underline indicate the best and second best.

| Method | Train: Qian's dataset \| Test: Test-a | | | | |
|---|---|---|---|---|---|
| | PSNR↑ | SSIM↑ | MUSIQ↑ | CLIPIQA+↑ | HyperIQA↑ |
| AGAN | 31.57 | 0.9023 | 70.52 | 0.6691 | 0.6650 |
| Histoformer | **33.06** | **0.9441** | 70.66 | 0.6530 | 0.6695 |
| DuRN | 31.24 | 0.9259 | 70.30 | 0.6433 | 0.6545 |
| DiffUR[3] | 31.66 | 0.9324 | **72.22** | **0.6857** | **0.7058** |

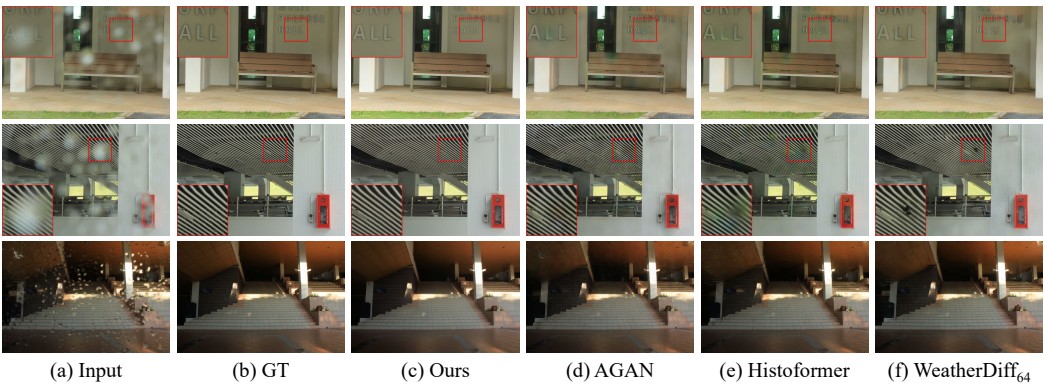

(a) Input    (b) GT    (c) Ours    (d) AGAN    (e) Histoformer    (f) WeatherDiff$_{64}$

Figure 15: Visual results of various methods on Test-a dataset. Please check and zoom in on screen for a better view.

