# OpenReview forum: "Unified Removal of Raindrops and Reflections: A New Benchmark and A Novel Pipeline"
_ICLR.cc/2026/Conference — ICLR 2026 Conference Withdrawn Submission_

### Official Review · Reviewer_vq1v · 2025-10-28

**Soundness:** 1
**Presentation:** 3
**Contribution:** 1
**Rating:** 2
**Confidence:** 4

**Summary:**

This paper proposes a new task of removing both rain and reflection at the same time. It proposes a new dataset that contains both raindrops and reflections, and a model, trained on the proposed dataset, for removing both raindrops and reflections from the input image.

**Strengths:**

According to this submission, the proposed model proposes state-of-the-art results on the proposed dataset, compared with prior raindrop removal methods, reflection removal methods and cascading methods.

**Weaknesses:**

- What is the synergy of combining the raindrop removal task with the reflection removal task? This is not intuitive to me, and needs to be discussed in the paper. Otherwise, one can simply combine any two existing tasks to become a new task, for example, raindrop removal with shadow removal to become a new task, reflection removal and shadow removal to become another new task, and perhaps combining all three tasks into yet another new task?

- While the comparison of the proposed method with two cascading methods (Histoformer+DAI and DAI+Histoformer) in Table 2 shows that the proposed method performs better, it does not mean that these results can generalize when other rain/reflection removal models are used. First, Histoformer is more like a general model, not a model designed specifically for raindrop removal. Second, the experiment is conducted on the proposed dataset only. A more important question, in my own opinion, is why your model performs better than the cascading methods. This is not discussed in the paper.

- Another important problem of this work is that the combination of the two existing tasks into a new task makes the problem scope very narrow - removing raindrops and reflections from images taken through a piece of glass during a rainy day with both raindrops and reflections on the glass.

- In terms of technical novelty, I am not sure why decomposing the pipeline into Restoration and Conditional Generation is novelty. I am unable to find any explanation of this in the submission.

**Questions:**

See my comments in the Weaknesses section.

---

### Official Review · Reviewer_jmmm · 2025-10-28

**Soundness:** 2
**Presentation:** 3
**Contribution:** 2
**Rating:** 4
**Confidence:** 4

**Summary:**

This paper introduces a new image restoration task: the Unified Removal of Raindrops and Reflections(UR$^3$). First, the authors construct a benchmark dataset called RDRF (RainDrop and ReFlection), collected using a specialized image acquisition platform to capture paired clean and degraded images. Then, they propose a diffusion-based pipeline, DiffUR$^3$, to address the DiffUR$^3$ task. The model first creates a preliminary restoration and then uses both this initial result and the original degraded image as dual conditions to guide a generative diffusion process, achieving state-of-the-art results on the new benchmark.

**Strengths:**

1. This work is the first to define and tackle the UR$^3$ task, which is of some importance in certain practical scenarios.
2. The experimental results are compelling.

**Weaknesses:**

1. The proposed RDRF dataset has some deficiencies:

- In the dataset, the raindrops are created by spraying water onto the glass surface, which may not fully capture the diverse characteristics and distribution of real-world raindrops.

- Compared to the training set (216 scenes with 9003 pairs), the test set is quite small (36 scenes with 83 image pairs).

2. The experiments are insufficient:

- The model complexity and inference time of the proposed DiffUR$^3$ should be reported, in comparison with other methods.

- The evaluation lacks a comparison against more general "all-in-one" or "composite degradation" restoration frameworks, such as [1] [2] [3] [4]:

  [1] Guo, Yu, et al. "Onerestore: A universal restoration framework for composite degradation." *European conference on computer vision*. Cham: Springer Nature Switzerland, 2024.

  [2] Zhou, Yingjie, et al. "Q-Agent: Quality-Driven Chain-of-Thought Image Restoration Agent through Robust Multimodal Large Language Model." *arXiv preprint arXiv:2504.07148* (2025).

  [3] Cao, Jin, Deyu Meng, and Xiangyong Cao. "Chain-of-Restoration: Multi-Task Image Restoration Models are Zero-Shot Step-by-Step Universal Image Restorers." *arXiv preprint arXiv:2410.08688* (2024).

  [4] Mao, Jiawei, et al. "AllRestorer: All-in-One Transformer for Image Restoration under Composite Degradations." *arXiv preprint arXiv:2411.10708* (2024).

**Questions:**

The authors should address the issues mentioned in the "Weaknesses".

---

### Official Review · Reviewer_46dV · 2025-10-31

**Soundness:** 3
**Presentation:** 3
**Contribution:** 2
**Rating:** 4
**Confidence:** 4

**Summary:**

This paper addresses the joint removal of raindrops and reflections by introducing a new dataset and a diffusion-based framework. The method adopts a two-stage design that effectively restores degraded images and improves visual quality. Experiments show clear advantages over existing deraining and reflection removal approaches.

**Strengths:**

(1) Novel problem and dataset contribution. This work identifies a clear research gap by jointly addressing raindrop and reflection removal, which are traditionally treated separately. The proposed RDRF dataset is well-designed, diverse, and of high resolution, making it a valuable benchmark for the community.

(2) Strong technical framework and convincing results. The DiffUR3 framework is well-motivated, integrating diffusion priors with a modular design. The proposed Modulate&Gate and Fidelity Encoder effectively enhance detail consistency and texture realism. Ablation studies and both objective and subjective evaluations solidly support the claimed improvements.

**Weaknesses:**

(1) Comparisons mainly involve single-degradation or re-trained baselines. Including recent all-in-one restoration or multi-condition diffusion methods would make the evaluation more comprehensive.
(2) Incremental gain mainly from encoder tuning. As shown in Table 1, the main performance boost appears to come from the Fidelity Encoder fine-tuning, while the Modulate&Gate module contributes relatively little improvement. Similar encoder-side refinements have already been explored in prior works such as StableSR and MPerceiver, raising concerns that the improvement is more due to enhanced VAE adaptation than to fundamentally novel architectural design.
(3) Lack of efficiency analysis. The method involves multiple heavy components (DRSformer, Stable Diffusion, Fidelity Encoder), but no runtime or memory analysis is reported, which limits understanding of its practical deployability.
(4) Insufficient generalization discussion. Although some in-the-wild results are provided, quantitative analysis on cross-domain or adverse lighting conditions is lacking. Failure cases or robustness discussion would strengthen the paper

**Questions:**

(1) How does the Modulate&Gate mechanism differ quantitatively from standard conditioning methods like SPADE (StableSR) or AdaIN , and can it generalize to other diffusion backbones?
(2)  Is the Fidelity Encoder trained jointly or separately from the diffusion branch, and how sensitive is performance to its training schedule? From Table 1, most of the performance gain seems to originate from the fine-tuning of the Fidelity Encoder, while the proposed Modulate&Gate module contributes only marginal improvement. Could the authors clarify why this happens, and how the encoder tuning differs from previous works such as StableSR or MPerceiver?
(3) How well does the method generalize to dynamic or outdoor scenes with changing lighting and raindrop motion?
(4) What is the inference time and resource cost of DiffUR3?

---

### Official Review · Reviewer_2Y2D · 2025-11-01

**Soundness:** 2
**Presentation:** 2
**Contribution:** 3
**Rating:** 4
**Confidence:** 5

**Summary:**

This paper proposes a new diffusion-based method to tackle a compounded image degradation problem of raindrops and reflection. The paper curated a new real-world image dataset that contains the raindrop&reflection degradation and ground truth. The dataset has almost 10K images including training and testing sets. Based on the collected dataset, the proposed method learns to remove the degradation effects from input images. The proposed method contains two stages. The first stage is a simple model that directly learns to remove the degradation effect and distortion. The second stage designs a modulate&gate modules that carefully extract the corresponding features from input images and output of stage 1. The features are used to guide the decode to restore from the diffusion U-nets. Extensive experiments show that the proposed method is the most effective compared with baseline methods on the new curated dataset.

**Strengths:**

1. This paper proposes a new high quality real-world dataset containing raindrops & reflections with ground truths. The dataset will have considerably high impact for raindrop/reflection removal community.
2. The quantitative results of the proposed method outperforms the baseline methods (either off-the-shelf or retrained).

**Weaknesses:**

1. Line 255, the explicit goal of the aim is not clearly explained. What does it mean by remove degradations without introducing new artifacts? how does the first stage achieve this goal? What are the simple degradations?
2. Line 273,  image reflection applies to all the region on the reflective surface. If the input images are fully covered by the reflective images, there will be no region clean out of the reflection effect.
3. Table 1, the PSNR score without Modulate&Gate is even higher than baseline 2. What is the purpose and usage of the modulate&gate module?
4. The proposed method has limited novelty in terms of resolving the compounded problem of single image reflection and raindrop image problem. The proposed hand-crafted network architecture does not show extensive effectiveness compared to existing methods. In addition, the intuitive theoretical reasoning is also not convincingly stated in the methodology section.

**Questions:**

The main concern of the paper is the novelty of the proposed method. The method description does not clearly reveal the insights of the network design and the method does not show superiority by adopting the designs. The authors are suggested to address those issues during rebuttal period.

---

### Note · Authors · 2025-11-13

I have read and agree with the venue's withdrawal policy on behalf of myself and my co-authors.